Construction of a high-density genetic map using specific-locus amplified fragment sequencing and quantitative trait loci analysis for tillering related traits in Psathyrostachys juncea perennial grass

Ma Yingmei 1
Chang Yudong 2
Li Zhen 2
Gao Zhiqi 2
Han Feng 2
Wang Yong 2
Yun Lan 2 3 nmg_yunlan@163.com
1 College of Desert Control Science and Engineering, Inner Mongolia Agricultural University , Hohhot , China
2 College of Grassland Science, Inner Mongolia Agricultural University , Hohhot , China
3 Key Laboratory of Grassland Resources of the Ministry of Education , Hohhot , China
Abd El-Moneim Diaa
Electronic publication date: 2024 Nov 6
Publication date: 2024
Volume: 12
Electronic Location ID: e18409
Received 2024 Aug 19; Accepted 2024 Oct 6
Copyright: © 2024 Ma et al.
Copyright year: 2024
Copyright holder: Ma et al.
License: This is an open access article distributed under the terms of the Creative Commons Attribution License, which permits unrestricted use, distribution, reproduction and adaptation in any medium and for any purpose provided that it is properly attributed. For attribution, the original author(s), title, publication source (PeerJ) and either DOI or URL of the article must be cited.
License URL: https://creativecommons.org/licenses/by/4.0/

Keywords: Russian wildrye, Phenotypic trait, Molecular map, Quantitative trait loci

Funding: National Natural Science Foundation of China 32371762 Natural Science Foundation of Inner Mongolia of China 2023ZD07 This research was funded by the National Natural Science Foundation of China, grant number 32371762 and the Key Project of the Natural Science Foundation of Inner Mongolia of China, grant number 2023ZD07. The funders had no role in study design, data collection and analysis, decision to publish, or preparation of the manuscript.

==============================
Background

Russian wildrye (RWR, Psathyrostachys juncea) is an outcrossing perennial grass that plays a crucial role in foragaing and rangeland restoration due to its tiller producing capabilities, nevertheless, a genetic map has yet to be constructed due to a shortage of efficient and reliable molecular markers. This also limits the identification, localization, and cloning of economically important traits related to tiller density during breeding.

Methods

Therefore, this study aimed to create a F1 mapping population with 147 individual lines and their two parents, which were selected based on varying tiller densities. We then used this mapping population to conduct specific-locus amplified fragment sequencing (SLAF-seq) to generate SLAF markers and discover single nucleotide polymorphisms (SNPs).

Results

Initially, we generated a total of 1,438.38 million pair-end reads with an average sequencing depth of 84.92 in the maternal line, 79.34 in the parental line, and 27.05 in each F1 individual line, respectively. Following the filtering of low-depth SLAF tags, a total of 558,344 high-quality SLAFs were identified. A total of 1,519,903 SNP markers were obtained, and 62,424 polymorphic SNPs were discovered. From these, 4,644 polymorphic SNPs were selected and used for the construction of a genetic map encompassing seven linkage groups. The genetic map spanned 1,416.60 cM with an average distance of 0.31 cM between adjacent markers. Comparative analysis between the seven linkage groups of RWR SLAF tag and the whole-genome sequences in barley (Hordeum vulgare L.) revealed homology values ranging from 17.5% to 34.6%, and the collinearity between the RWR linkage groups and the barley homology groups ranged from 0.6787 to 0.9234, with an average value of 0.8158. Additionally, 143 significant quantitative trait locus (QTLs) with Logarithm of Odds (LOD) value greater than 2.5 for five tiller related traits were detected using three consecutive years of phenotypic trait data from the F1 population, further verifying the map’s reliability.

Introduction

Disturbance of rangeland results in the loss of vegetation cover and the intrusion of undesirable plant species, exacerbating rangeland degradation by increasing competition among plants for limited soil nutrients (Pyke, Wirth & Beyers, 2013; Robins, Bushman & West, 2017). To protect and restore these rangelands, robust and productive plant species are essential. Consequently, Russian wildrye (RWR; Psathyrostachys juncea), native to the steppe and desert regions of Russia, Mongolia, China, and Central Asia (Xiong et al., 2020), Thus, the selection and breeding of robust tiller producing Russian wildrye, to enhance its utility in forage and rangeland restoration, are critical for improving this species. Over past years of cultivation, Russian wildrye has been recognized as a robust tillering type of grass. Fertile tillers are important for both seed yield (Wang et al., 2011) and forage yield in Russian wildrye (Domingues et al., 2021), this phenomenon was also noted during field evaluation trials conducted over the past several years. Research has shown that diploid Russian wildrye shows lower seed vigor and seedling growth as compared to tetraploid. The capacity to produce tillers from primordial meristems or buds is influenced by both environmental factors and genetic makeup (Kumar et al., 2021). Li et al. (2022b) elucidated 10 candidate genes associated with seedling and tillering in Russian wildrye through transcriptome analysis and they found high expression levels of genes involved in the biosynthesis of strigolactones (SL), indoleacetic acid (IAA) and cytokinins (CTK) during the tillering stage. To address this tiller production issue, a previous study crossed plants with sparse and dense tiller growing types using two diploids (2n = 2x = 14), aiming to select lines with excellent tiller production and to facilitate map-based cloning of genes controlling tiller (Gao et al., 2022). Furthermore, a high-density genetic map is essential for utilizing and pinpointing the genes that control tillering. For most perennial grass and legume species with strict heterogamous pollination habits, it is necessary to use individual plants with heterozygous genetic backgrounds for crosses and deriving a mapping population. F1 populations derived from two heterozygous parents, which show significant trait segregation, have been ideal for mapping populations and have been used in QTLs detection for many species, such as alfalfa (Zhang et al., 2020). In this study, we used two parent plants with genetically distant and F1 progenies for specific-locus amplified fragment sequencing (SLAF-seq) analysis and QTLs detection. We selected the closest diploid relatives of barley (Hordeum vulgare) as references, which have published whole genome sequence information (Mascher et al., 2017), for initial linkage group construction and marker alignment analysis. Population size affects the results of QTL detection, Lv et al. (2021) conducted a QTL detection using a RIL population of 198 wheat plants and identified two QTL loci determining plant height. Che et al. (2024) conducted a QTL detection on a CP population of 113 agropyron gaertn plants, which resulted in the detection of 28 QTL loci on seven linkage groups (LGs).

A high-density genetic map is a valuable resource for genomic analysis. The lack of rich and reliable molecular markers, along with the shortage of high-density linkage maps, poses significant challenges in identifying and locating genes that control specific traits. A critical prerequisite for constructing a high density genetic map is the availability of a large number of polymorphic markers. Specific-locus amplified fragment sequencing (SLAF-seq) is a cost-effective technique that employs efficient high-throughput sequencing technology to identify single nucleotide polymorphism (SNP) markers by reducing the complexity of high-quality reference genome libraries (Sun et al., 2013). This technique has been applied in common carp (Cyprinus carpio; Sun et al., 2013), sesame (Sesamum indicum; Zhang et al., 2013), soybean (Glycine max; Qi et al., 2014; Li et al., 2014), Ume (Prunus mume; Zhang et al., 2015), cucumber (Cucumis sativus; Wei et al., 2014; Xu et al., 2015), orchard grass (Dactylis glomerata; Xiong et al. 2020), and Carpinus oblongifolia (Li et al., 2022a) for constructing high density genetic maps.

To date, research on Russian wildrye has focused on collection, protection, and utilization of this exceptional germplasm resource, as well as improvements in disease resistance and tissue culture techniques (Yu & Zhang, 2010). However, the genetic architecture of RWR tillering traits remains largely unexplored. Therefore, this newly reported high-density genetic map of Russian wildrye will serve as a valuable tool for gene mapping and QTL detection for tillering related genes. It will also provide references for marker assisted breeding, candidate gene identification, and draft genome assembly of RWR.

Materials and Methods

Construction of mapping population and DNA extraction

Portions of this text were previously published as part of a preprint (Ma et al., 2020). The mapping population comprised two Psathyrostachys juncea (Fisch.) Nevski (synonym Elymus junceus Fischer) parents and 147 F1 individual lines, derived from a cross between a robust tillering line as the male parent (PI 549118), provided by the National Plant Germplasm System, USA, and a weak tillering line as the female parent (CF 005038), obtained from the National Medium-term Gene Bank of Forage Germplasm, China. The experimental site was located in Hohhot, Inner Mongolia China (111.41°E, 40.48°N), with altitude of 1,050 m, average temperature of 6.7 °C, and precipitation of about 400 mm per year. The soil is a sandy calcium soil with a pH value of 7.0–7.5. The male parent line was characterized by dense nutritional and robust fertile tiller production while the female line was noted for sparse tiller production. Additionally, the female parent line had significant advantages in seed-related traits such as longer spike length, a higher spikelet number per spike, and a higher seed setting rate. The procedures for parent selection, cross making, and offsprings cultivation are described in Gao et al. (2022). Prior to flowering, the female parents were bag-isolated to prevent pollen from non-target plant. Male pollen was collected in the full flowering stage for artificial bagging pollination, with each combination being pollinated three to five times. The hybrid seeds of each female parent were harvested. We selected combinations with high seed setting rate (30%) and hybrid populations with sufficient offspring (more than 100 plants) and cultivated them in the April of 2018 at a natural environment forage experiment station located at the Inner Mongolia Agricultural University in Saihan District, Hohhot, Inner Mongolia, China (111.41°E, 40.48°N). The experimental plots were 30 cm2 in size with plant spacing in rows of 50 cm × 50 cm.

We collected fresh leaves from the 147 individuals and their parents, and then we froze them in liquid nitrogen for genomic DNA preparation. Total genomic DNA was extracted utlizing a plant genome DNA extraction kit (Tiangen Biotech, Co., Ltd, Beijing, China). We estimated DNA concentration and quality with a NanoDrop-2000 spectrophotometer (Thermo Fisher Scientific, Waltham, MA, USA) and by electrophoresis in 1.0% agarose gels with 5 K marker (Trans Gen Biotech, Beijing, China). Based on these tests, the concentration and quality of all samples were qualified and could be used for constructing the SLAF library.

SLAF library construction and high-throughput sequencing

A pre-designed SLAF experiment was conducted, employing a control to predict the results of enzyme digestion and assess its efficacy. SLAF Library construction and sequencing procedure showed in Fig. S1. Rice (Oryza sativa japonica) was chosen as the control based on its genome size (382 M) and GC content, with sequences available for download from http://rice.plantbiology.msu.edu/. The selection and digestion of the optimum enzyme were based on its restriction site, which was located in regions with less repetitive sequences, and the enzyme fragment was uniformly distributed across the genome. Upon comparison of the enzyme digestion effectiveness with the reference genome, we found that 92.73% of the control was normally digested, indicating the DNA was effectively digested. The experimental system was further optimized based on the fragments produced by enzyme digestion. Finally, two enzymes were selected to completely digest the genomic DNA of the qualified RWR parents. Fragments digested by Rsal enzymes have more evenly distribution across the genome, the enzyme Rsal (New England Biolabs, NEB, Ipswich, MA, USA) was used to digest the genomic DNA, Following complete digestion, the ends of different fragments were repaired to create blunt-ended DNA and the 5′ends were phosphorylated. We subsequently added a single nucleotide adenine (A) overhang to the digested fragments using Klenow Fragment (3′→ 5′ exo–) (New England Biolabs, NEB, Ipswich, MA, USA) and dATP at 37 °C. Duplex tag-labeled sequencing adapters (PAGE-purified; Life Technologies, Carlsbad, CA, USA) were then ligated to the A-tailed fragments using T4 DNA ligase, enabling parallel sample sequencing for high-throughput sequencing. We then performed polymerase chain reaction (PCR) using diluted restriction-ligation DNA samples, dNTP, Q5® High-Fidelity DNA Polymerase and PCR primers (Forward primer: 5′-AATGATACGGCGACCACCGA-3′, reverse primer: 5′-CAAGCAGAAGACGGCATACG-3′) (PAGE-purified, Life Technologies) to enrich the DNA products. The PCR products were then purified using AgencourtAMPure XP beads (Beckman Coulter, High Wycombe, UK) and pooled. The pooled samples were then separated on a 2% agarose gel electrophoresis (120 V, 60 min), and fragments ranging from 314–344 bp, with indexes and adaptors, were targeted, excised, and purified using a QIAquick gel extraction kit (Qiagen, Hilden, Germany). The gel-purified products were then diluted to equal concentration for sequencing. Pair-end sequencing (Each end 125-bp) was performed on an Illumina HiSeq 2500 system (Illumina, Inc.; San Diego, CA, USA) according to the manufacturer’s instructions.

Sequence data grouping and genotyping

SLAF marker identification and genotyping were conducted using the methodology outlined in Sun et al. (2013). Initially, low-quality reads (quality score < 20e) were filtered out, and then raw reads were sorted to each progeny according to duplex barcode sequences. Subsequently, barcodes and the terminal 5-bp positions were trimmed from each high-quality read. The cleaned reads from the same sample were then aligned to the barley (Hordeum vulgare) genome sequence using SOAP software (Liu et al., 2014). Sequences that mapped to the same position were categorized as a single SLAF locus (Davey et al., 2013). Alleles of each SLAF locus were then defined based on parental reads with sequence depth greater than 82.13 fold while alleles for each offspring were determined using the reads with sequence depth >27.05. In diploid species, a single SLAF locus can contain up to four genotypes; therefore, SLAF loci with more than four alleles were considered repetitive and discarded. Only SLAFs exhibiting two to four alleles were recognized as polymorphic and considered potential markers. All polymorphism SLAFs loci were genotyped, ensuring consistency between the parental and offspring SNP loci. The marker code of the polymorphic SLAFs were analyzed based on the cross-pollination population type which includes one segregation types (aa × bb).

Genotype scoring was conducted using a Bayesian approach to enhance the genotyping quality (Sun et al., 2013). Initially, the posteriori conditional probability was calculated based on the coverage of each allele and the number of single nucleotide polymorphisms. Then, a genotyping quality score derived from this probability was used to select for qualified markers for further analysis (Sun et al., 2013). Low-quality markers were counted for each individual and the worse marker per individual was removed during the dynamic process. The process was stopped once the average genotype quality scores of all SLAF markers reached the cutoff value. High-quality SLAF markers intended for genetic mapping required average sequence depths less than 3-fold in each progeny and less than 4-fold in the parents. Screening for markers with genotypes covering at least 70% of individuals in all offspring, markers with more than 30% missing data were also removed. Finally, a chi-square test was conducted to assess segregation distortion, so markers with significant segregation distortion (P < 0.05) were initially excluded from the map construction, but they were then added later as accessory markers.

Linkage map construction

Marker loci were initially grouped into linkage groups (LGs) based on their locations on the barley (Hordeum vulgare) genome. Next, the modified logarithm of odds (MLOD) scores between markers were calculated to further validate the robustness of the markers for each LGs. Markers with MLOD scores less than five were excluded before ordering. To facilitate the efficient construction of a high-density and high-quality map, a newly developed HighMap strategy was employed to orangize the SLAF markers and correct genotyping errors within LGs (Kozich et al., 2013). First, recombinant frequencies and LOD scores were calculated by two-point analysis which was used to infer linkage phases. The process then, incorporated enhanced Gibbs sampling, spatial sampling, and simulated annealing algorithms in an iterative procedure for marker ordering (Li et al., 2008; Arabidopsis Genome Initiative, 2000). In the first phase of the ordering procedure, SLAF markers were selected using spatial sampling. A marker was randomly chosen with priority given to test crosses, and markers with a recombination rate smaller than a given sampling radius were removed from the marker set. Simulated annealing was subsequently applied to identify the optimal map order. The summation of adjacent recombination fractions was calculated as illustrated by Liu et al. (2014). The annealing system continued until, in a number of successive steps, the newly generated map order was rejected. Blocked Gibbs sampling was utlized to estimate the multipoint recombination frequencies of the parents after determining the optimal map order of sample markers (Liu et al., 2014). The revised recombination frequencies were then used to integrate the two parental maps, optimizing the map order in the next cycle of simulated annealing. Once a stable map order was obtained after three to four cycles, the next map building round was initiated. A subset of currently unmapped markers was selected and added to the previous sample, with a reduced sample radius. The mapping algorithm was repeated until all the markers were mapped appropriately. The error correction strategy of SMOOTH was implmented based on the parental contribution of genotypes (Li & Durbin, 2009), and a k-nearest neighbor algorithm was employed to impute missing genotypes (Stam, 2009). Skewed markers were then incorporated into the map using a multipoint method of maximum likelihood. Map distances were estimated using the Kosambi mapping function (Tang, Sezen & Paterson, 2010).

Determination of genomic homology and collinearity between Russian wildrye and barley

The 4,644 SLAF markers used in the construction of the Russian wildrye genetic map were compared against the latest whole-genome sequences of barley using the BLAST function. For each mapped marker, the 100 bp sequence flanking either side were used to compare with the barley genome. In BLAST, the default e value of 1e−10 represent that the comparison results are considered statistically significant. The total number of BLAST hits meeting the minimum threshold of an e value < 1e−10 were recorded. Hits were considered accurate for a chromosome if the e-value of the best hit for any marker with multiple hits on the same chromosome was at least 1e−5 lower than the next best hit. Once the SLAF markers were mapped to the barley genome, those with significant hits underwent further analysis for their homologous group assignments, and the number of markers with such assignments were counted. Genome homology, expressed as a percentage, between the linkage groups of Russian wildrye and barley genomes was calculated by dividing the number of significant hits of one Russian wildrye LG on a given chromosome of barley by the sum of significant hits on a given chromosome of barley. The collinearity between the linkage group of Russian wildrye and the corresponding homologous chromosome in barley was assessed using the Spearman coefficient (Wang et al., 2020a). Additionally, a chi-square test with a Bonferroni adjustment for multiple tests was conductec to determine if the observed values differed significantly.

Traits observation and QTL analysis

The F1 population and the two parental lines used for map construction were also utilized for collecting phenotype data and conducting QTL analysis. We conducted normal distribution test and phenotypic trait test for five traits in three growing seasons using the software origin2021 (OriginLab, Northampton, MA, USA). QTL mapping was performed using a high-density genetic linkage map alongside phenotypic data related to tiller characteristics. Traits such as plant height (PH), tiller number (TN), base bundle diameter (BD), spike length (SL) and spike width (SW) were measured in the F1 population for three consecutive years. We used MapQTL 6.0 (developed by Kyazma Company) to estimate the phenotypic variation of each significant QTL (Raymond et al., 2014). QTLs were identified using the IM interval mapping method and the LOD threshold set at 0.90. A permutation test was, repeated 1,000 times, established the a logarithm of odds (LOD) threshold, and a LOD score larger than 2.5 cutoff was used to identify significant QTL (Broman et al., 2003). The nomenclature of QTL loci is according to Xue et al. (2022). Markers located at or flanked with the peak LOD value of a QTL were recognized as QTL-associated markers.

Results

Library construction and SLAF sequencing of mapping population

Genomic DNA from 147 F1 progeny individual lines and two parental lines were used for specific-locus amplified fragment sequencing analysis. Oryza sativa L. spp. japonica, var Nipponbare served as as a control to determine the effectiveness of the enzyme digestion scheme. Sequence quality control involved comparisons with the reference genome using the SOAP (short oligonucleotide alignment program; Li et al., 2008). The enzyme digestion efficiency for the control was 92.73% (Table 1). The efficiency of pair end comparison of the control to this database construction was 88.83%, indicating effective enzyme digestion. After the construction of the SLAF library and high-throughput sequencing, we obtained 1,438.38 M pair-end reads with lengths from 314–344 bp. The guanine-cytosine (GC) content of RWR was 43.71% and Q30 ratio (a quality score of 30) was 95.31% (Table 2).

Table 1 Enzyme digestion results for construction of specific-locus amplified fragment (SLAF) library.

Name of the content	Number of the content	
Digesting enzyme	Rsal	
Enzyme digestion rate of control	92.73%	
Pair end comparison efficiency of control	88.83%	
Length of enzyme digestion	314–344 bp	
Number of reads after enzyme digestion	451,235	
Clean Reads	1,438.38 M reads	
Average number of Q30	95.31%	
Average content of GC	43.71	
Average number of SLAF reads	558,344	
Sequencing depth of parental reads	82.13X	
Sequencing depth of offspring reads	27.05X	
Number of linkage groups	7	
Number of SLAF markers for map construction	4,644	
Total marker distance	1,416.60	
Average marker distance	0.31 cM	

Table 2 Detailed statistics of specific-locus amplified fragment sequencing (SLAF-seq).

Sample	Total reads	Q30 percentage (%)	GC percentage (% )	
Paternal	32,236,395	95.70	44.32	
Maternal	30,241,864	95.93	44.30	
Offspring	9,359,885	95.30	43.70	
Control	710,325	95.35	39.19	
Total	1,438,381,311	95.31	43.71	

SNPs discovery and genotyping

SNP detection was performed using the GATK (developed by Broad Institute, Cambridge, MA, USA) toolkit based on the sequence positioning reference genome. Local realignment was performed around the sites of the insertions and deletions to correct alignment errors caused by these alterations, as well as variation in SNP, indel, and mutations. In the maternal line, 357,751 SNPs were generated from a total of 1,519,903 SNPs, with a 11.84% hetero ratio (the proportion of different alleles carried by an individual at a single locus). In the paternal line, 352,039 SNPs were generated from 1,519,903 SNPs with a 15.18% hetero ratio. For the F1 population, 269,953 SNPs out of 1,519,903 SNPs were generated, with a hetero ratio of 9.91% (Table 3).

Table 3 Detailed information for single nucleotide polymorphism (SNP) discovery and hetero ratio.

Sample	Total SNP	SNP number	Hetero ratio	
Paternal	1,519,903	357,751	11.84%	
Maternal	1,519,903	352,039	11.18%	
Offspring	1,519,903	269,953	9.91%	

After filtering outlow-depth SLAF tags, we detected a total of 558,344 high-quality SLAFs. 1,519,903 SNP markers were obtained, and 62,424 polymorphic SNPs were identified. Of these, 31,867 polymorphic SNPs were selected for genetic map construction. In order to facilitate the follow-up genetic analysis, we had to encode the genotype of polymorphic SNPs. As Russian wildrye is a cross-pollination (CP) species, the parental genotypes were not pure aa or bb. Following the general two-allele coding rule in genetics, the genotypes for parents and their possible offspring are presented in Table 4.

Table 4 Genotype for parents and their possible combinations of offspring.

Type	Paternal genotype	Maternal genotype	Offspring genotype	
ab × cd	ab	cd	ac, ad, bc, bd, --	
ef × eg	ef	eg	ee, ef, eg, fg, --	
ab × cc	ab	cc	ac, bc, --	
cc × ab	cc	ab	ac, bc, --	
hk × hk	hk	hk	hh, hk, kk, --	
lm × ll	lm	ll	lm, ll, --	
nn × np	nn	np	nn, np, --	
aa × bb	aa	bb	F2 (aa, ab, bb), RIL/DH (aa, bb)--	

Out of the 1,519,903 polymorphic SNPs, 62,424 were categorized into eight segregation patterns (ab × cd, ef × eg, ab × cc, cc × ab, hk × hk, lm × ll, nn × np, and aa × bb) following a genotype encoding rule (Fig. 1). Due to the CP population, polymorphic markers other than aa × bb type were selected as effective markers suitable for this population. To ensure the quality of the genetic map, polymorphic SNPs were filtered out if the sequencing depth of the parents was less than 4×, the SNP depth of the offspring was less than 3×, and the selected genotypes covered less than 70% of the individual markers in all offspring, as well as in the cases of partial separations (P < 0.01). Finally, 4,778 markers were successfully selected for genetic map construction (Table 5).

Figure 1 Basic statistics of specific-locus amplified fragment (SLAF) markers suitable for the population under different genotype combinations.

Table 5 Single nucleotide polymorphism (SNP) markers that successfully selected for genetic map construction under different genotype combinations.

Genotype combinations	SNP number	Percentage	
ef × eg	3	0.06 %	
hk × hk	200	4.19 %	
lm × ll	2,362	49.43 %	
nn × np	2,213	46.32 %	
Total	4,778	100.00 %	

High-throughput linkage map construction

These 4,778 SNPs were organized into seven linkage groups (LGs) based on initial positioning relative to the reference genome of barley. The MLOD values between pairs of SNPs were calculated (Stam, 1993), and SNPs with MLOD values less than five were removed.

Following linkage analysis, the 4,644 markers were mapped onto seven linkage groups by estimating the genetic distance using HighMap (Liu et al., 2014; Fig. 2; Table S1). The map spanned a total of 1,416.60 cM with an average marker distance of 0.31 cM. On average, each LG had 663 SNPs. The genetic distances across the seven LGs ranged from 182.98 cM (LG5) to 225.35 cM (LG2), with the average marker distance varying from 0.26 cM (LG2) to 0.35 cM (LG3). The largest gap within the seven LGs ranged from 4.26 cM (LG7) to 12.31 cM (LG3) (Table 6).

Figure 2 Genetic maps of Russia wildrye in seven linkage groups using specific-locus amplified fragment (SLAF) markers.

Table 6 Distribution of specific-locus amplified fragment (SLAF) markers on map of seven linkage groups of diploid Russian wildrye.

Linkage group	Total marker	Total distance (cM)	Average distance (cM)	Max gap (cM)	Gap < 5 cM	
1	666	211.58	0.32	11.41	99.25	
2	883	225.35	0.26	8.11	99.77	
3	607	215.00	0.35	12.31	99.17	
4	755	203.42	0.27	7.98	99.34	
5	464	182.98	0.40	10.73	99.14	
6	650	194.88	0.30	5.03	99.85	
7	619	183.39	0.30	4.26	100.00	
Total	4,644	1,416.60	0.31	12.31	99.50	

Evaluation of the genetic map

To evaluate the quality of the constructed map, a chi square test (P < 0.01) was applied to the polymorphic markers. The average sequencing depth ensured the accuracy of molecular markers (Table 7). Furthermore, the average integrity of the markers on each individual within the mapping population was 99.99% which also affirmed the accuracy of map genotyping (Fig. 3).

Table 7 The basic statistics for sequencing depth of specific-locus amplified fragment (SLAF) markers used for map construction.

Sample	Marker number	Total depth	Average depth	
Paternal	4,644	394,370	84.92	
Maternal	4,644	368,443	79.34	
Offspring	4,313	117,302	27.05	

Figure 3 The average integrity of the markers on each individual of the mapping population used for map genotyping.

The x-axis indicates the number of F1 individuals. The y-axis indicates marker coverage.

Based on the genetic structure analysis of each individual line from the F1 population, the origins of each individual across all linkage groups were consistent, and the incidence of potential double crossover sites was maintained below 3%, indicating the high quality of genetic map (Fig. 4). The genetic map is essentially a multi-point recombination analysis. The smaller the distance between markers, the smaller the recombination rate is. By analyzing the recombination between markers and their surrounding markers, potential error makers can be filtered out from the genetic map. Consequently, a heat map was generated using pair-wise recombination values for the 4,644 SNPs. It demonstrated that recombination between adjacent markers on each linkage group of the project was robust. With increasing distance, the linkage relationship between markers and distant markers decreased, indicating that markers in LGs are well ordered (Fig. 5).

Figure 4 Genetic structure analysis was performed on each individual from the F1 population to ensure the origin of each individual used for the map population.

Note: Each row represents a marker, which is arranged according to its position on the linkage group. Each column represents one chromosome in a sample, green represents from the maternal parent, blue represents the paternal parent, white represents undetermined, and gray represents missing. The location where the color of the same column changes is the location where the reorganization event occurs.

Figure 5 The linkage relationship between distant markers to reveal recombinations between markers and the surrounding markers.

Note: Each row and column are markers in the map order. Each small square represents the recombination rate between the two markers. The change of color from yellow to red to purple represents the change of recombination rate from small to large. The closer the marker is, the smaller the recombination rate is, and the closer the color is to yellow. The farther the marker is, the greater the recombination rate is, and the closer it is to purple.

Genomic homology and collinearity between Russian wildrye and barley

Since RWR (Psathyrostachys juncea) is a diploid member of the Triticeae taxonomic group, we selected the genome of barley, a diploid of the same group with a published whole genome sequence, for alignment analysis. Based on the 100 bp sequence flanking either side of the mapped SLAF markers in Russian wildrye linkage group, we conducted a BLAST search against the whole genome sequence data of barley to compare their linkage positions (blast_onmap_Hordeum_vulgare.final.posi). Of the 60,994 hits on the barley genome searched by the 4,644 mapped SLAF markers of Russian wildrye, 2,278 provided unambiguous hits that aligned with the barley homology groups (HGs). The overall homology (degree of similarity of molecular sequences between species) between Russian wildrye and barley was approximately 49.05%. Homology estimate using mapped 4,644 SLAF tags, the highest homology between Russian wildrye Linkage groups and barley HGs varied from 17.5% to 34.6% (Fig. 6; Table 8). Comparisons between the LGs of Russian wildrye and chromosomes of barley showed that LG1 was aligned to Chromosome 5, LG2 was aligned to Chromosome 7, LG3 was aligned to Chromosome 6, LG4 was aligned to Chromosome 2, LG5 was aligned to Chromosome 3, LG6 was aligned to Chromosome 1, and LG7 was aligned to Chromosome 4, respectively. Collinearity between Russian wildrye LGs and barley HGs, assessed by the Spearman coefficient, ranged from 0.6787 to 0.9234 individually, with an averaged value of 0.8158 in which three of seven alignments were negatively correlated (Table 9; Fig. 6). Due to the very limited number of markers used to estimate homology, the overall prediction of homology was relatively low. However, it is still possible to compare the relative homology levels between LGs and chromosomes.

Figure 6 BLAST search against whole genome sequence data of barley to compare their linkage positions with Russian wildrye.

Table 8 Comparison of genomic homology of Russian wildrye and barley based on specific-locus amplified fragment (SLAF) markers.

	Total of significant hits on		
Russian wildrye LG1	Russian wildrye LG2	Russian wildrye LG3	Russian wildrye LG4	Russian wildrye LG5	Russian wildrye LG6	Russian wildrye LG7	Sum	
Barley Chr1	36	52	30	31	11	93	16	269	
Barley Chr2	33	44	55	118	27	34	43	354	
Barley Chr3	45	66	32	45	60	42	53	343	
Barley Chr4	35	49	39	26	23	44	62	278	
Barley Chr5	100	63	32	53	30	32	61	371	
Barley Chr6	43	42	77	40	36	27	36	301	
Barley Chr7	61	105	39	54	33	39	31	362	
Barley Chr1 homology	13.4%	19.3%	11.2%	11.5%	4.1%	34.6%	5.9%		
Barley Chr2 homology	9.3%	12.4%	15.5%	33.3%	7.6%	9.6%	12.1%		
Barley Chr3 homology	13.1%	19.2%	9.3%	13.1%	17.5%	12.2%	15.5%		
Barley Chr4 homology	12.6%	17.6%	14.0%	9.4%	8.3%	15.8%	22.3%		
Barley Chr5 homology	27.0%	17.0%	8.6%	14.3%	8.1%	8.6%	16.4%		
Barley Chr6 homology	14.3%	14.0%	25.6%	13.3%	12.0%	9.0%	12.0%		
Barley Chr7 homology	16.9%	29.0%	10.8%	14.9%	9.1%	10.8%	8.6%		
Note:

Homology = Total number of significant hits on a given chromosome of Russian wildrye/Sum of markers on a given chromosome of Barley.

Table 9 Collinearity between specific-locus amplified fragment (SLAF) markers of Russian wildrye and barley genomes.

SLAF markers in LGs of
Russian wildrye	Spearman correlation coefficient	
Barley Chr1	Barley Chr2	Barley Chr3	Barley Chr4	Barley Chr5	Barley Chr6	Barley Chr7	
Russian wildrye Chr1-1	0.8960*	0.9071*	0.7740	0.8210*	0.8191*	0.9733*	0.5947	
Russian wildrye Chr1-2	0.8008	0.9889***	0.7508	0.5357	0.7992*	0.8704	0.8307	
Russian wildrye Chr1-3	0.8660	–	–	–	–	–	0.6333	
Russian wildrye Chr1-4	0.9703**	–	–	–	–	–	0.8528	
Russian wildrye Chr1-5	0.9110*	–	–	–	–	–	0.6409	
Russian wildrye Chr1-6	0.8013	–	0.7035	–	–	0.6236	0.6463	
Russian wildrye Chr2-1	–	–	0.7670*	–	0.6237	–	–	
Russian wildrye Chr2-2	–	0.7742	0.5071	–	0.7467	–	–	
Russian wildrye Chr2-3	–	0.5960	0.9918*	–	0.6937	0.8830	–	
Russian wildrye Chr2-4	0.9496*	0.8660	0.6438	0.8488*	0.8171*	0.6735	0.6454	
Russian wildrye Chr3-1	–	–	0.6408	–	–	–	–	
Russian wildrye Chr3-2	–	–	0.6867	–	0.4237	–	0.9128	
Russian wildrye Chr3-3	0.9045	0.6064	0.5399	0.9266	0.6736	0.8441*	0.5071	
Russian wildrye Chr3-4	0.8586	0.4472	0.9918*	0.9045	–	–	–	
Russian wildrye Chr3-5	–	–	0.7188	–	–	–	0.7434	
Russian wildrye Chr4-1	–	–	0.7720	0.9068*	0.8947*	–	–	
Russian wildrye Chr4-2	0.9703**	0.8291*	0.8013	0.9919*	0.7848	0.8932	0.8588*	
Russian wildrye Chr4-3	–	–	0.6318	0.6735	–	–	–	
Russian wildrye Chr4-4	0.9071*	0.9889***	0.7740	0.5300	0.7559	0.8588*	0.6347	
Russian wildrye Chr4-5	0.9487	0.5000	0.5991	0.9459	0.7696	0.6804	0.9324*	
Russian wildrye Chr5-1	–	–	–	–	0.6523	–	–	
Russian wildrye Chr5-2	–	–	–	–	0.6523	–	–	
Russian wildrye Chr5-3	0.8191*	0.5965	0.7508	0.9129	0.9068*	0.9709**	0.7382*	
Russian wildrye Chr5-4	0.7992*	0.5834	0.7238	0.8393	0.8947*	0.9708**	0.7391*	
Russian wildrye Chr5-5	–	–	–	–	0.8528	0.5744	–	
Russian wildrye Chr5-6	–	–	–	–	0.7035	–	–	
Russian wildrye Chr5-7	0.6333	0.6463	0.9733*	0.6867	0.7720	0.6498	0.6064	
Russian wildrye Chr5-8	–	–	–	–	–	0.821*	–	
Russian wildrye Chr5-9	0.7670*	0.6099	0.7229	0.5356	0.8488*	0.8655	0.6804	
Russian wildrye Chr6-1	0.9045	0.8704	0.5071	0.7395*	0.9401**	0.9419	0.8713	
Russian wildrye Chr6-2	0.8660	0.8214*	0.7404	0.8807	0.8540*	0.6438	0.6820	
Russian wildrye Chr6-3	0.6438	0.8528	0.7395	0.8079*	0.7734*	0.7708*	0.6438	
Russian wildrye Chr7-1	0.8215*	0.7461*	0.7285	0.9500*	0.5745	0.7035	0.7433	
Russian wildrye Chr7-2	–	–	0.6734	–	–	–	0.8255	
Russian wildrye Chr7-3	–	–	–	–	0.5960	0.8660	0.6134	
Russian wildrye Chr7-4	0.5158	0.5411	0.5079	0.5071	0.5399	0.7708*	0.6438	
Russian wildrye Chr7-5	0.7404	0.8210*	0.6408	0.6937	0.9266	0.8441*	0.8488*	
Russian wildrye Chr7-6	–	–	–	–	–	–	0.5960	
Russian wildrye Chr7-7	–	–	–	–	–	–	0.5000	
Russian wildrye Chr7-8	–	–	–	–	–	–	0.5399	
Russian wildrye Chr7-9	–	–	–	–	–	–	0.5356	
Russian wildrye Chr7-10	0.5356	0.5005	0.5773	0.6522	0.8214	0.9657**	0.9627**	
Russian wildrye Chr7-11	0.5773	0.6916	0.7461*	0.9656*	0.9988***	0.8454	0.7404	
Russian wildrye Chr7-12	0.4958	0.5773	0.6804	0.7285	0.9627*	0.8310	0.9274	
Russian wildrye Chr7-13	0.5947	0.8830	0.9459	0.6736	0.6317	0.6099	0.9401*	
Russian wildrye Chr7-14	0.5947	0.8830	0.9459	0.6736	0.6317	0.9813*	0.8454	
Russian wildrye Chr7-15	–	–	–	–	–	–	0.6522	
Russian wildrye Chr7-16	–	–	–	–	–	–	0.7461	
Russian wildrye Chr7-17	0.8306	0.7467	–	–	0.7695	–	0.9499*	
Mean	0.7285	0.6722	0.6939	0.7235	0.6863	0.7728	0.6822	
Note:

* Significant correlation at the P < 0.05 level.

** Highly significant correlation at the P < 0.01 level.

*** Highly significant correlation at the P < 0.001 level.

Tillering related trait QTLs detection

After three consecutive years of phenotypic evaluation and QTL detection, significant QTLs related to five traits: plant height (PH), tiller number (TN), base bundle diameter (BD), spike length (SL) and spike width (SW) have been identified. We conducted a normal distribution tests and phenotypic trait tests on the five traits across three growth seasons. Furthermore, histogram also displayed a distribution trend where higher values were concentrated in the middle and lower values on both ends. These results suggest that the five traits in the F1 population are approximately normally distributed and are ideal traits for QTL detection (Fig. 7, Table 10). We used a LOD value of 2.5 as the critical threshold for QTL analysis of five tiller related traits using the IM model of MapQTL. Our QTL analysis identified a total of 143 loci associated with five tillering-related traits across the genome (LG1-LG6) over 3 years. Importantly, four QTLs for plant height (PH) on LG5 were consistently detected in 2022 and 2023, suggesting the presence of stable genetic regions influencing this trait. This high level of reproducibility strengthens the confidence in these QTLs. The stringent filtering criteria employed during marker selection likely contributed to the identification of these robust QTL associations. including Marker1346 at 129.11 cM, Marker 11066 at 129.44 cM, Marker 3254 at 129.79 cM, and Marker 3251 at the 130.48 cM position, although the LOD value for these four QTLs in 2022 were a bit low (2.06 to 2.12). Four SL QTL loci on LG2 (Marker 6455 at 80.21 cM, Marker 6975 at 83.45 cM, Marker 4630 at 86.49 cM and Marker 28194 at 92.51 cM) were repetitive detected in 2021 and 2023, with LOD values in 2023 (2.57~3.46) higher than those in 2021 (2.68~2.80). The LOD values for significant association between markers and traits ranged from 2.35 to 4.63, and the phenotypic variation explained (PVE) by these markers ranged from 7.1% to 13.5% (Table S2; Fig. 8).

Figure 7 Phenotype frequency distributions for plant height, tiller number, base bundle diameter, spike length and spike width in the F1 population of Russian wildrye.

Table 10 Phenotypic analysis of tiller-related traits of Psathyrostachys juncea parents and F1 populations.

Trait	Year	Average value	Maximum value	Minimum value	Coefficient of variation	Standard deviation	
PH	2021	109.72	145.00	39.00	15.82	17.35	
	2022	126.79	176.00	25.00	18.55	23.52	
	2023	114.94	176.00	71.00	13.14	15.10	
TN	2021	155.72	234.00	52.00	25.85	40.26	
	2022	159.92	284.00	34.00	34.30	54.84	
	2023	110.26	178.00	42.00	22.55	24.87	
BD	2021	42.16	60.00	19.00	20.02	8.24	
	2022	48.48	88.00	21.00	24.11	11.69	
	2023	64.77	138.00	25.00	33.39	21.62	
SL	2021	9.12	12.80	3.96	16.34	1.49	
	2022	10.32	15.30	5.90	17.70	1.83	
	2023	12.34	16.20	8.90	11.34	1.40	
SW	2021	0.54	0.95	0.16	24.27	0.13	
	2022	0.54	0.82	0.12	30.00	0.16	
	2023	0.71	0.90	0.40	18.23	0.13	

Figure 8 QTL mapping for plant height (A), tiller number (B), base bundle diameter (C), spike width (D) and spike length (E), in Russian wildrye, each panel shows the QTL distribution of a target trait.

In each panel, the X-axis represents the linkage group and the Y. Each panel shows the QTL distribution of a target trait. In each panel, the X-axis represents the linkage group and the Y-axis shows the corresponding LOD values (red lines).

Discussion

Genetic control of tiller density in Russian wildrye become possible based on high-density genetic map

Plant tillering is crucial for yield formation and viability, influenced by both genes and environmental factors. For example, in rice (Oryza sativa L.), tillering is integral to yield formation where tillers originate from the establishment of axillary meristem and further developed via axillary bud formation and bud outgrowth (Hussien et al., 2014). Two genes of Monoculm1 (MOC1) and Lax panicle1 (LAX1; Komatsu et al., 2003; Li et al., 2003) have been identified as critical for the initiation and formation of axillary buds while the bud outgrowth is regulated by genetic factors, environmental cues, and hormonal signals (Hussien et al., 2014). Research also indicates that short term drought stress significant reducers tiller number whereas long term drought stress results in axillary bud number reduction in tall fescue (Festuca elata Keng ex E. Alexeev; Zhuang, Wang & Huang, 2017). A total of 3 days of low temperature (12 °C) results in slight stress on rice tillering while 10 days of low temperature (15 °C) significantly decreases rice tiller number (Liu et al., 2019).

In this study, we found that the male and female parents used to establish the mapping population are genetically and geographically distinct, with each exhibiting unique morphological traits (Gao et al., 2022). Given the harsh conditions of the rangeland, plant materials must be capable of effectively reproducing tillers to thrive and persist in such environments. Thus, we hypothesized that the traits associated with tillering in Russian wildrye might be regulated by specific genes, so constructing a genetic map focused on genes related to tiller density could be a strategic approach in breeding this species for high-yield forage production and active rangeland restoration. However, the specific genetic location of these tiller related genes in the DNA sequences of Russian wildrye is currently unknown, due to the species’ low genetic diversity and the absence of high-density genetic map. Currently, only conventional markers such as RAPD (Wei, Campbell & Wang, 1997), AFLP (Xiong et al., 2020), and SSR (Li et al., 2022b) are available for genome studies in Russian wildrye. Based on a study of genetic diversity and structure, the genetic diversity among Russian wildrye germplasm is low, resulting in limited polymorphisms for the existing markers. SLAF-seq serves as one of the strategies for large scale SNP discovery and genotyping in species lacking reference genome information (Sun et al., 2013). Therefore, this study aimed to construct a high-density genetic map using the progenies (F1) and parents’ materials via SLAF-seq. Following the construction of the SLAF library and high-throughput sequencing, we obtained 1,438.38 M pair-end reads with lengths of 314–344 bp. Within those large scale SLAF makers, we identified many SNPs for the construction of the high-density genetic linkage map. Due to the CP population, polymorphic markers of types other than aa × bb type were selected as effective markers for the population. Specifically, a total of 1,519,903 SNP markers were obtained, among which 62,424 polymorphic SNPs were discovered; 31,867 of these were selected for construction of the genetic map via classifying them into eight segregation patterns (ab × cd, ef × eg, ab × cc, cc × ab, hk × hk, lm × ll, nn × np, and aa × bb) following a genotype encoding rule. This resulting high-density molecular map spanned 1,416.6 cM with an average marker distance of 0.31 cM. This first high-density genetic map developed for genes related to tillering traits across seven linkage groups of Russian wildrye will facilitate the identification and map-based cloning of these genes. The use of this map will lay the foundation for gene mapping, map-based gene cloning, and marker-assisted selection in Russian wildrye.

High-density genetic map and QTL detection in Russian wildrye is reliable

To explore novel genomic information, QTL mapping and gene discovery of economically important agronomic traits in plant species via the construction of a high-density genetic map is a crucial tool for such research (Luo et al., 2020). This study is the first report genetic maps of the tiller density related genes in diploid Russian wildrye. Evaluating the quality of the genetic map requires sufficient molecular markers, an adequate population size, and highly efficient genotyping approaches (Luo et al., 2020). In this study, we evaluated the quality of the genetic map with chi square test, high marker integrity, low genetic differences between individuals, and appropriate recombination based on heat maps. Initially, due to the cross-pollination (CP) species of Russian wildrye, of the 1,519,903 polymorphic SNPs, 62,424 were classified into eight segregation patterns (ab × cd, ef × eg, ab × cc, cc × ab, hk × hk, lm × ll, nn × np, and aa × bb) following a genotype encoding rule for parents and their possible offspring based on a two-allele coding rule. Polymorphic SNPs in LGs were filtered out if the sequencing depth of parents was less than 4x, the SNP depth of offspring was less than 3x, and the selected genotypes covered less than 70% of the individual markers in all offspring, alongside significant partial separations (P < 0.01; Table 4). In the end, 4,778 markers were successfully selected for the construction of the genetic map (Table 5).

The relationship between recombination and markers within a LG could be used to determine the potential ordering errors among the markers (Wang et al., 2020b). A strong linkage relationship between two adjoining markers across the seven LGs was evident from the heat map analysis which confirms the order of markers in the LGs as the linkage strength decreased as the distance between two markers increased (Fig. 5). Additionally, marker integrity and accuracy were validated through offspring integrity analysis (Fig. 3). Evaluation of the chromosome monomer origin of each progeny across all linkage groups revealed that the origin of larger segments in each individual remained consistent, and the frequency of double exchange was below 3%, affirming the reliability of the genotyping results and the linkage map (Fig. 4). This newly constructed genetic map related to tillering traits of Russian wildrye will provide valuable references for the assembly and evaluation of future genome sequencing efforts in Russian wildrye and other diploid or polyploid perennial grass species with genes related to tiller density. Tillering-related traits play an important role in forage yield and seed production of forage crops. While QTL mapping for plant branching and tillering traits has been reported in annual crops such as Wheat and Rice (Cai et al., 2024; Lei et al., 2018), little is known about perennial forage grass species. In this study, the distinct genetic backgrounds and phenotypic traits of the two parents plants resulted in high heterosis in tillering related traits in the F1 population (Gao et al., 2022). The same F1 population used for genetic mapping were also used for phenotypic traits evaluation and QTL analysis.

Genome relationships between Russian wildrye and barley based on SLAF markers

Chromosome karyotyping (Hsiao, Wang & Dewey, 1986), chromosome pairing in hybrids (Wang, 1989, 1992), genomic in situ hybridization (Schwarzacher et al., 1989), isozymes, molecular markers (Wei & Wang, 1995; Linc et al., 2017), comparative genomics (Devos & Gale, 2000), and DNA sequences of genes or gene families represented as phylogenetic trees or dendrograms (Peng, Hu & Yang, 2015; Li et al., 2015) have been employed for investigating genome relationships. However, quantitatively determining homology between the Russian wildrye genome and the barley genome is challenging due to the lack of intergeneric hybrids of diploid species. With the advent of high-throughput sequencing, genome-wide marker mapping and sequencing now enable a quantitative assessment of the genome relationships between two plant genomes (Li et al., 2015; The International Wheat Genome Sequencing Consortium (IWGSC) et al., 2018). For example, among 423,611 wheat expressed sequence tags (ESTs), 101,299 (23.9%) were commonly expressed in rice, maize, and barley (Tang et al., 2006). Another study (Bilgic et al., 2007) determined that the homology between barley and wheat was approximately 44.8% and the synteny was 63.2%. Since there is no published genome sequence information for the RWR, it is difficult to carry out genetic precision targeting. It is feasible to use species of the same ploidy, with the same number of chromosomes, and closely related species as reference. In this study, the SLAF sequences of Russian wildrye genome shared approximately 49.05% homology with barley whole-genome sequences, and the highest homology ranged from 17.5% to 34.6% on seven barley chromosomes (Table 8). Russian wildrye exhibited slightly lower collinearity, as indicated by the spearman correlation coefficient, between LGs and HGs of these two species, which ranged from 0.6787 to 0.9234, especially in LG5 and LG7 (Table 9; Fig. 6). Although Russian wildrye showed a closer relationship with barley than wheat and rice in traditional plant taxonomic system, the genetic relationship between Russian wildrye and barley spans different genera and is not closely linked, which might explain this low level of collinearity. Thus, the further the relationship indicated by the collinearity analysis, the fewer shared gene families with barley are present in Russian wildrye. This discrepancy could be due to the shortcomings of mapping without a known referencing genome. Additionally, the collinearity between Russian wildrye and barely might be underestimated due to the SLAF sequence used in analysis, which may not capture the complete genomic information of Russian wildrye. Although more markers should be employed to enhance the density of the Russian wildrye genetic map relating to tillers, this high-density genetic map constructed using SLAF makers still serves as a valuable tool for researching the genetic control, localization, and cloning of tiller related genes in Russian wildrye.

Application of the genetic linkage map in QTLs detection

In this study, the distinct genetic backgrounds and phenotypic traits of the two parents plants resulted in high heterosis in tillering related traits in the F1 population (Gao et al., 2022). The same F1 population used for genetic mapping were also used for phenotypic traits evaluation and QTL analysis. We conducted QTL mapping for five tillering-related traits for over three years. Significant QTLs were identified for five of these traits using an LOD threshold of 2.5, including PH, TN, BD, SL and SW. The distribution of these five traits in the F1 population were more closely aligned with a normal distribution (Fig. 7). Four PH QTL loci on LG5 and four SL QTL loci on LG2 were repeatedly detected across more than two growth seasons. Notably, some loci were found to be associated with more than two traits, such as marker 7909 loci in LG3 associated with both TN and BD in 2022, marker 60075 loci in LG6 associated with BD in 2021 and SW in 2022, marker 34206 loci in LG2 associated with both PH and TN in 2023, marker 60937 and marker 26435 loci in LG4 associated with SL in 2021 and PH in 2023. Marker 26435 and marker 60937, which are closely located in LG4, demonstrated a high explanatory variation for the SL trait (Table S2). These marker loci will serve as a focus for candidate gene screening in future studies. Meanwhile, based on the results of QTL detection, we found that the locus for PH trait with physical position 16.8 had the same physical position as the previous study on QTL mapping for wheat PH (Lv et al., 2021). The significant and repeated detection of QTLs for tiller-related traits using this P.juncea high-density genetic linkage map further validates the reliability of the map and provide a theoretical basis for marker-assisted breeding of other perennial grasses. At the same time, we expect to screening candidate genes controlling tillering-related traits in subsequent work.

Conclusion

A high-density genetic map of tiller related genes was constructed for an F1 population using SLAF-seq analysis, representing the highest density molecular linkage map to date for the cross-pollination (CP) species of Russian wildrye (Psathyrostachys juncea). The detection of significant QTLs for five tiller related traits further substantiates the map’s reliability. The QTLs detected in this study require further validation to investigate their stability through fine-mapping. In the future, generating BC1 populations and increasing the sample size for out-cross species is a possible approach. This study provides a theoretical basis for understanding the genetic basis of tillering in perennial grasses. Utilizing this genetic map will provide a solid foundation for advancing research in tiller related gene mapping, map-based gene cloning, and marker-assisted breeding of Russian wildrye, ultimately enhancing forage production.

Supplemental Information

Supplemental Information 1 Experimental flow of SLAF-seq.

Supplemental Information 2 Genetic map.

Supplemental Information 3 Quantitative trait locus.

The authors would like to thank Beijing Bemac Biologicals for providing the bioinformatic analyses. We would also like to thank all the reviewers for their constructive comments and suggestions on this article.

Additional Information and Declarations

Competing Interests

Author Contributions

Data Availability

The authors declare that they have no competing interests.

Yingmei Ma performed the experiments, analyzed the data, prepared figures and/or tables, authored or reviewed drafts of the article, and approved the final draft.

Yudong Chang performed the experiments, analyzed the data, prepared figures and/or tables, authored or reviewed drafts of the article, and approved the final draft.

Zhen Li analyzed the data, prepared figures and/or tables, and approved the final draft.

Zhiqi Gao analyzed the data, authored or reviewed drafts of the article, and approved the final draft.

Feng Han performed the experiments, authored or reviewed drafts of the article, and approved the final draft.

Yong Wang performed the experiments, prepared figures and/or tables, and approved the final draft.

Lan Yun conceived and designed the experiments, authored or reviewed drafts of the article, and approved the final draft.

The following information was supplied regarding data availability:

The sequence data of Russian wildrye (Psathyrostachys juncea) used for genetic map construction in this study is available at NCBI: PRJNA935976.

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
