# Peer review of "Construction of a high-density genetic map using specific-locus amplified fragment sequencing and quantitative trait loci analysis for tillering related traits in Psathyrostachys juncea perennial grass"

_PeerJ, doi:10.7717/peerj.18409_

## Round 0.1 · original submission · Minor Revisions

Dear Authors

The manuscript cannot be accepted for publication in its current form. It needs a minor revision before publication. The authors are invited to revise the paper considering all the suggestions made by the reviewers. Please note that the requested changes are required for publication.

With Thanks

Reviewer 1 ·

Basic reporting

No comment

Experimental design

The study demonstrates an acceptable experimental design.

Validity of the findings

No comment

Additional comments

The study represents a significant contribution to the understanding of the RWR genome and provides a valuable resource for future research on this important forage species. The genetic map and identified QTLs can be used to accelerate breeding efforts and develop cultivars with improved tiller density and other desirable traits. However, further research is needed to validate the identified QTLs and investigate their underlying molecular mechanisms.
-Comments and Suggestions for Authors
- Abstract
-The abstract provides a clear and concise overview of the study. It effectively outlines the research question, methodology, and key findings.

- Introduction
- While the introduction effectively outlines the problem, the specific research question could be stated more explicitly.
-The introduction effectively outlines the significance of Russian wildrye in rangeland restoration and highlights the challenges associated with tiller production.
- The introduction could provide a more direct link to the specific methods used in the study, such as the choice of parent plants and the SLAF-seq analysis.
- The introduction might benefit from a brief acknowledgment of potential limitations or challenges associated with the research, such as the availability of reference genomes or the complexity of tiller development.
- Materials & Methods
-This section provides a detailed description of the methods used in this research.
- Consider adding a sentence or two explaining the purpose of bag-isolating the female parent (line 113).
- Briefly explain the criteria for selecting "combinations with high seed setting rate and hybrid populations with sufficient offspring" (line 116).
-Some sections could be slightly condensed. For example, consider summarizing the specific software used for sequence alignment and marker filtering (lines 160-163) instead of listing them all.
-While some parameter choices are explained (e.g., LOD threshold), consider briefly mentioning the rationale behind others, such as the minimum e-value for BLAST hits (line 217).

- Results
- This section provides a detailed description of the results obtained from your research.
-Some sections could be condensed by focusing on key findings and omitting procedural details (e.g., lines 279-281 describing marker removal).
- Briefly mention how specific methods (e.g., filtering criteria for markers) contributed to the results (e.g., high-quality genetic map).
- Consider explaining the significance of metrics like "heterozygosity ratio" (line 259) and "homology" (line 313) for readers who might not be familiar with these terms.
- While you describe various aspects of the results, consider emphasizing the most important findings at the beginning of each subsection.
Here's an example of how you could revise a section to highlight key findings and connect them to methods:
Original:
A total of 39 QTL loci associated with PH, 19 QTL loci related to TN, 29 QTL loci related to BD, 21 QTL loci related to SL, and 35 QTL loci related to SW were detected in at least one year, distributed across LG1 to LG6. Notably, four PH QTL loci on LG5 were detected repeatedly in consecutive years 2022 and 2023... (lines 332-335)
Revised:
Our QTL analysis identified a total of 143 loci associated with five tillering-related traits across the genome (LG1-LG6) over three years. Importantly, four QTLs for plant height (PH) on LG5 were consistently detected in 2022 and 2023, suggesting the presence of stable genetic regions influencing this trait. This high level of reproducibility strengthens the confidence in these QTLs. The stringent filtering criteria employed during marker selection (lines 273-276) likely contributed to the identification of these robust QTL associations.

- Discussion
- The discussion section provides a strong synthesis of the study's findings and their implications.
- Remove the subtitles in discussion section.
- While the discussion mentions the influence of environmental factors on tillering, it could benefit from a more in-depth exploration of how these factors might interact with the genetic factors identified in the study.
- The discussion could provide a more specific outlook on future research directions based on the study's findings. For example, the authors could suggest potential candidate genes for further investigation.

- Conclusion
The conclusion could be strengthened by elaborating on the potential implications of this research beyond forage production, such as its contribution to understanding the genetic basis of tillering in perennial grasses.

Reviewer 2 ·

Basic reporting

Thank you for considering me for reviewing the manuscript "Construction of a high-density genetic map using SNPs makers and QTL analysis for tillering related traits in Psathyrostachys juncea perennial grass". The manuscript represents a significant contribution to genetic analysis of Psathyrostachys juncea. This study has potential to be a valuable resource for researchers and breeders in the field of forage crops.

Suggestion:
Conduct a thorough English language edit of the manuscript to correct grammatical errors and improve clarity. Summarize redundant sentences and focus on highlighting the important aspects of the study.

The title could be improved by considering using specific-locus amplified fragment sequencing for better specificity.

The abstract provides a concise overview of the research, including objectives, methods, results, and conclusions. It effectively highlights the significance of the study in the context of rangeland restoration and forage improvement. It could benefit from a more explicit statement on the practical implications of the findings, particularly how they can be applied in breeding programs.

Include relevant keywords that are not presented in the title and abstract to improve searchability.

The introduction needs reorganization to reduce redundancy, particularly in the sections presenting the importance of tillering trait. Provide a solid background on the role of Psathyrostachys juncea in forage production and clearly emphasize the need for genetic mapping to enhance breeding efforts. The rationale for selecting barley as a reference genome requires clarification to justify its use. Additionally, it is crucial to focus on the lack of a high-density genetic map for P. juncea, highlighting the gap this study aims to fill. The research objective should be stated clearly, emphasizing the novelty of using SLAF-seq in this context. Use updated citations; present references from 1963, 1987, and 1997. The introduction is not recommended.

The methodology is described in detail including parent selection, DNA extraction, and sequencing processes. However, the phenotyping trail is not clear needs more information on soil of the experimental site, environmental conditions of three growing seasons, agronomic practices, …. . Consider including a diagram summarizing the key steps in the SLAF-seq process to improve readability. The choice of enzyme could be justified further by presenting its relevance to the genome of P. juncea. It would be beneficial to briefly explain the significance of the chosen quality control thresholds e.g., 30% missing data.

The results section presents the construction of a high-density genetic map, detailing the number of markers, linkage groups, and map coverage. The section of the collinearity between P. juncea and barley could be expanded to explore biological significance. Descriptive statistics, including the mean, minimum, maximum, standard deviation, and coefficient of variation for the studied agronomic traits observed in the assessed genotypes, could be summarized in a table.

The discussion could search deeper into the biological implications of the collinearity findings, particularly how they might inform future research on the genetic basis of tillering and functional validation of the identified QTLs. Using barley as a reference genome should be discussed better. Also, future research directions including the potential for marker-assisted selection, should be discussed in detail. Provide more context on the identified QTLs, such as their potential impact on breeding programs or their relevance to previous studies.

The conclusion is brief could be improved by reinforcing the practical implications of the research by exploring how the breeders could exploit the genetic map and QTLs

The references section requires careful revision to ensure consistency with the journal guidelines. Make sure all in-text citations are correctly formatted. Ensure that the journal names are consistently abbreviated across all references. For instance, some are abbreviated as "Herb J. Syst. Bot" in line 510, "J. Range Management" in line 512, "Can. J. Plant Pathol." in line 518, ……. Other journal names are currently not abbreviated, such as "Theoretical and Applied Genetics" in line 522, "Molecular Ecology" in line 532, and "Forest Ecology and Management" in line 538,…..
In line 522, the scientific name "Triticum aestivum" should be formatted in italics along with all other scientific names throughout the manuscript.

Experimental design

Experimental design is appropriate

Validity of the findings

The findings presented in the manuscript are valid, supported by robust data and sound statistical analysis.

Reviewer 3 ·

Basic reporting

The manuscript titled "Construction of a high-density genetic map using SNP markers and QTL analysis for tillering-related traits in Psathyrostachys juncea perennial grass" presents valuable research with potential contributions to the field of genetics and plant breeding. However, the manuscript currently exhibits several deficiencies that need to be addressed to enhance its clarity, accuracy, and overall quality. Notably, the selection of keywords is redundant with the title, which limits the manuscript's discoverability in academic databases. The writing contains instances of unclear phrasing and long, complex sentences, which detract from the reader's focus and comprehension. Additionally, there are typographical errors that require correction."
The "Material and Methods" section lacks sufficient detail, particularly in outlining the experimental design, soil type, and the specific dates of the experiments. Furthermore, the SLAF marker identification procedure is described without providing crucial data. Discrepancies in the number of traits reported for QTL mapping, as well as a lack of appropriate QTL nomenclature, also need to be reconciled to ensure consistency and adherence to standard practices. The visual presentation of Figures are suboptimal, with small font sizes on the axes that impede readability, Figures must be of high resolution at least 600 dpi. The discussion section of this manuscript effectively outlines the significance of the high-density genetic map developed for Russian wildrye and its implications for understanding tiller-related traits. However, there are areas where further refinement could enhance its impact and clarity. The potential applications of this genetic map in breeding programs for Russian wildrye are briefly mentioned, but a more detailed exploration of how these findings could be operationalized in breeding strategies would be valuable. Additionally, the discussion could further address the limitations of the current study, particularly in relation to the SLAF-seq approach and the challenges of mapping without a reference genome. The authors discuss the genetic map and its utility in QTL mapping separately. Integrating these discussions to emphasize how the map directly contributes to the identification of QTLs for tillering traits would strengthen the narrative. This integration would also help to more clearly illustrate the practical implications of the genetic map, particularly in relation to improving forage yield and seed production in Russian wildrye. While the discussion references relevant studies, it could more critically compare the findings of this study with those of previous research. For instance, the authors note the detection of significant QTLs but do not sufficiently compare these QTLs with those identified in similar studies in other species. A more critical comparison could highlight the novelty of the findings and the potential for cross-species insights. The discussion occasionally strays into overly technical details that may be better suited for the results section or supplementary materials. Streamlining these sections and focusing on the broader implications of the findings would enhance the readability and accessibility of the discussion. Additionally, the discussion could be more concise, with an emphasis on key takeaways rather than extensive recapitulation of the results. The conclusion of this manuscript effectively summarizes the key achievements of the study. However, the conclusion could be strengthened by addressing a few critical points. First, the conclusion would benefit from a more explicit discussion on the broader implications of the findings, such as how this genetic map compares to existing maps in terms of accuracy, resolution, and applicability to other related species. Additionally, the potential challenges or limitations encountered in the study, as well as future research directions, could be briefly mentioned to provide a more balanced and forward-looking perspective. Furthermore, while the conclusion touches on the application of the genetic map in advancing research and breeding programs, it could be more specific about the next steps required for translating these findings into practical outcomes, such as the identification of candidate genes or the development of specific markers for breeding programs. A discussion on the potential for integrating this genetic map with other genomic tools or resources could also enhance the conclusion's depth and relevance. Addressing these issues will significantly improve the manuscript's quality, making it more rigorous, accessible, and impactful within the scientific community.

Specific Comments:
Keywords: The keywords selected for this manuscript are identical to those used in the title, which reduces their effectiveness in enhancing discoverability in academic databases. It is recommended that the authors provide 5-7 unique keywords that accurately reflect the core themes of the study, while avoiding redundancy with the title.

Line No. 52: The phrase "some breeding lines exhibited weak tillering and experienced died back at the center" lacks clarity and precision. It is suggested to rephrase this sentence to better convey the intended meaning, such as "some breeding lines exhibited weak tillering and suffered from central dieback."

Sentence Structure: Long, complex sentences can detract from the reader's focus and make the manuscript difficult to follow. This issue is particularly noticeable in the sentences found in Line Nos. 59-62, 108-140, and 94-97, among others. It is advised that these sentences be revised for clarity and conciseness to improve overall readability.

Line No. 61-62: There is a typographical error in the spelling of "indole acetic acid." Please correct it to ensure accuracy.

Material and Methods: Introductory Paragraph: The section would benefit from an introductory paragraph that clearly outlines the experimental design, including details such as soil type, fertility status, and the specific dates of the experiment across the three years of study. This will provide context and clarity for the subsequent methods.
SLAF Marker Identification (Line 156-161): The methodology section mentions the SLAF marker identification and genotyping procedure but lacks specific details. The total number of reads, as well as the number of filtered/discarded reads, should be included to give a complete picture of the data processing.
In Results (Line 313-315): The manuscript mentions homology percentages between Russian wildrye and barley without clarifying how these two homologies differ. This distinction should be made explicit, with a more detailed explanation of the homology data provided.
Normal Distribution (Line 327-328): While the text states that normal distribution curves for five traits were symmetrically distributed, it does not specify the statistical tool or software used for this analysis. This information should be included in the methodology, along with a more thorough discussion of these traits in both the methods and results sections.
Trait Details (Line No. 454-455): The manuscript states that QTL mapping was conducted for ten tillering-related traits, but the methods section only describes five traits. It is essential to reconcile this discrepancy by accurately listing all measured traits in the materials and methods section. Additionally, the software and tools used to construct Figure 7 should be specified.
QTL Nomenclature:
There is an established system for naming QTLs, yet the QTLs identified in this study have not been named according to these conventions. It is recommended that the authors assign appropriate nomenclature to the QTLs to align with standard practices and enhance the clarity and utility of the findings.
Line No. 467: “The significant and repeated detection og QTLs for tiller-related traits” correct og to of.
The current presentation of Figure 2 lacks optimal clarity, which could hinder the reader's ability to fully comprehend the data. To enhance the visibility and overall readability of the figure, it is recommended to increase the font size of both the x-axis and y-axis titles. This will ensure that the labels are easily distinguishable and clear when the figure is viewed in both print and digital formats. Additionally, the tick sizes along both axes should be enlarged to improve their legibility. These adjustments will contribute to a more effective visual communication of the data, thereby aiding in the overall interpretation and impact of the figure.

Experimental design

Experimental design needs further clarifications. For required details please check basic reporting.

Validity of the findings

The findings are promising.

---

## Round 0.2 · Minor Revisions

Dear Authors

The manuscript still needs a minor revision as per the final comments of R3.

With Thanks

Reviewer 1 ·

Basic reporting

No comment

Experimental design

No comment

Validity of the findings

No comment

Additional comments

The authors have made the changes I suggested in the last review. I recommend its publication in this journal.

Reviewer 2 ·

Basic reporting

I would like to thank the authors for their thorough revisions and thoughtful responses to previous comments. After reviewing the revised manuscript, I am pleased to see that all of my concerns have been adequately addressed. The authors have made the necessary changes and provided clear explanations where appropriate. I have no further comments or suggestions, and I believe the manuscript is now suitable for publication.

Experimental design

Experimental design is appropriate

Validity of the findings

The findings presented in the manuscript are valid, supported by robust data and sound statistical analysis.

Reviewer 3 ·

Basic reporting

I am pleased to note that the authors have carefully addressed all the queries in the revised manuscript. However, there are still some typographical, tense, and grammatical errors that the authors must thoroughly address during the proofreading process. Once these issues are resolved, the manuscript is suitable for publication in PeerJ.

Experimental design

Please provide details on the plot size, plant-to-plant spacing, row-to-row spacing, and replication information. Additionally, specify whether the experimental design was based on a randomized complete block design (RCBD) or another design.

Validity of the findings

Findings are promising.

---

## Round 0.3 · accepted · Accept

Dear Authors,

I am pleased to inform you that the manuscript has improved and can be accepted for publication.

Congratulations on accepting your manuscript, and thank you for your interest in submitting your work to PeerJ.

With Thanks